# Neonatal valproic acid exposure produces altered gyrification related to increased parvalbumin-immunopositive neuron density with thickened sulcal floors

**Kazuhiko Sawada**[1]*, **Shiori Kamiya**[1], **Ichio Aoki**[2]*

**1** Department of Nutrition, Faculty of Medical and Health Sciences, Tsukuba International University, Tsuchiura, Ibaraki, Japan, **2** Department of Molecular Imaging and Theranostics, NIRS, National Institutes for Quantum and Radiological Science and Technology (QST), Chib, Japan

\* k-sawada@tius.ac.jp (KS); aoki.ichio@qst.go.jp (IA)

**Data Availability Statement:** All relevant data are within the manuscript and its Supporting Information files.

## Abstract

Valproic acid (VPA) treatment is associated with autism spectrum disorder in humans, and ferrets can be used as a model to test this; so far, it is not known whether ferrets react to developmental VPA exposure with gyrencephalic abnormalities. The current study characterized gyrification abnormalities in ferrets following VPA exposure during neonatal periods, corresponding to the late stage of cortical neurogenesis as well as the early stage of sulcogyrogenesis. Ferret pups received intraperitoneal VPA injections (200 µg/g of body weight) on postnatal days (PD) 6 and 7. BrdU was administered simultaneously at the last VPA injection. *Ex vivo* MRI-based morphometry demonstrated significantly lower gyrification index (GI) throughout the cortex in VPA-treated ferrets (1.265 ± 0.027) than in control ferrets (1.327 ± 0.018) on PD 20, when primary sulcogyrogenesis is complete. VPA-treated ferrets showed significantly smaller sulcal-GIs in the rostral suprasylvian sulcus and splenial sulcus but a larger lateral sulcus surface area than control ferrets. The floor cortex of the inner stratum of both the rostral suprasylvian and splenial sulci and the outer stratum of the lateral sulcus showed a relatively prominent expansion. Parvalbumin-positive neuron density was significantly greater in the expanded cortical strata of sulcal floors in VPA-treated ferrets, regardless of the BrdU-labeled status. Thus, VPA exposure during the late stage of cortical neurogenesis may alter gyrification, primarily in the frontal and parietotemporal cortical divisions. Altered gyrification may thicken the outer or inner stratum of the cerebral cortex by increasing parvalbumin-positive neuron density.

## Introduction

Sulcogyral development has been documented in the cerebrum of several mammalian species, including humans [1], baboons [2], cynomolgus monkeys [3–7], common marmosets [8], bovine [9], dogs [10], and ferrets [11–13]. The cerebral cortex can be roughly differentiated into four types of surface morphology. Ferrets, humans, and non-human primates have a

**Funding:** This work is supported by Japan Society for the Promotion of Science (JSPS) KAKENHI (grant no.15K08144 to Kazuhiko Sawada; grant no. 17H00860 to Ichio Aoki), a grant from Japan Agency for Medical Research and Development (grant no. 17dm0107066h to Ichio Aoki) and the Center of Innovation (COI) Program of Japanese Science and Technology Agency (JST) (grain no. JPMJCE1305 to Ichio Aoki). The funders had no role in study design, data collection and analysis, decision to publish, or preparation of the manuscript.

**Competing interests:** The authors have declared that no competing interests exist.

relatively large cerebral cortex with high gyrification [14]. Many reports have documented cortical gyrification abnormalities in human neurodevelopmental disorders like autism spectrum disorder (ASD) [15–20], schizophrenia [21, 22], and bipolar disorder [23, 24]. It has been speculated that these gyrification abnormalities contribute to atypical cortical connectivity [18, 21, 25, 26]. However, gyrification abnormalities have also been associated with cortical thickness [17, 19, 25, 27] and the number of cortical neurons [28].

Ferrets are gyrencephalic animals that undergo sulcogyrogenesis during the first three weeks after birth [11–13]. Genetic manipulations [29–31] and prenatal methylazoxymethanol exposure [32] have been shown to cause gyrification abnormalities in ferrets. Valproic acid (VPA), a widely used antiepileptic/anticonvulsant drug that inhibits histone deacetylases 1 and 2 [33, 34], may also result in abnormal morphology. Developmental exposure to VPA can cause teratological effects in the brain. This can produce ASD-like behavioral phenotypes, including reduced social interaction in rodents [35–38], and cortical dysgenesis with increasing neuron numbers and/or thickening in humans [39], mice [40–43], and rats [44]. Notably, VPA exposure during the brain growth spurt period produces ASD-like social behaviors in ferrets [45].

The current study characterized gyrification abnormalities in ferrets following VPA exposure on postnatal days (PDs) 6 and 7. This time point corresponds to the late stage of cortical neurogenesis [46] and the early stage of sulcogyrogenesis [12, 13]. Self-renewable neuronal stem cells, differentiated as basal radial glia (bRG) or outer radial glia, are abundant in the outer subventricular zones of the developing cerebral cortex [47] and contribute to the vast expansion of the cerebral cortex in mammalian species, including ferrets [48]. VPA-induced gyrification abnormalities were analyzed based on neuron density and cortical stratum thickness to observe mechanical compression in the cortical floor of the primary sulci.

## Materials and methods

### Animals

Eight male ferret pups delivered naturally from six pregnant ferrets were purchased from Japan SLC (Hamamatsu, Japan). The pups were reared with lactating mothers (4–6 pups/ mother) in stainless-steel cages (80 cm × 50 cm × 35 cm) maintained at 21.5°C ± 2.5°C under 12-h artificial illumination in the Facility of Animal Breeding, Nakaizu Laboratory, Japan SLC. All lactating mothers were given a pellet diet (High Density Ferret Diet 5L14, PMI Feeds, Inc., St. Louis, Mo.) and tap water *ad libitum*.

Four ferret pups received intraperitoneal VPA injections (200 μg/0.01 mL/g body weight) on PDs 6 and 7. These days correspond to the late stage of cortical neurogenesis [46] and the early stage of sulcogyrogenesis [12, 13]. 5-Bromo-2′-deoxyuridine (BrdU) (30 μg/0.01 mL/g body weight, Sigma-Aldrich) was simultaneously injected with the last VPA administration. The four control pups received BrdU on PD 7. All ferrets were reared until complete primary sulcogyrogenesis at PD 20 [12, 13]. They were then deeply anesthetized under ~2% isoflurane gas, perfused with PBS, and then perfused with 4% paraformaldehyde (PFA) in PBS. The body and brain weights of VPA-treated and control ferret on PD 20 used in this study are shown in S1 Table. The brain weight did not differ between two groups, although the body weight was significantly greater in VPA-treated ferrets than in control ferrets.

All study procedures were performed in accordance with the National Institutes of Health's (NIH) Guide for the Care and Use of Laboratory Animals and approved by the Institutional Animal Care and Use Committee of Tsukuba International University. All procedures attempted to minimize the number and suffering of animals used.

## MRI measurements

MRI measurements were performed in accordance with our previously published work [13]. Briefly, a 7.0-T MRI system (magnet: 400 mm inner diameter bore, Kobelco and Jastec, Kobe, Japan; console: AVANCE-I, Bruker BioSpin, Ettlingen, Germany) was used to acquire three-dimensional (3D) MRIs from the fixed ferret brains. 3D MRIs of the entire brain were obtained using the rapid acquisition with relaxation enhancement (RARE) sequence. The following parameters were used: repetition time (TR) = 300 ms, echo time (TE) = 9.6 ms (effective TE = 19.2 ms), RARE factor = 4, field of view = $32 \times 32 \times 40$ mm$^3$, acquisition matrix = $256 \times 256 \times 256$, voxel size = $125 \times 125 \times 156$ μm$^3$, number of acquisitions = 2, and total scan time = 2 h 43 min 50 s.

## MRI-based morphometry

Cerebral cortical volume, cortical surface area, fronto-occipital (FO)-length, and cerebral hemisphere width were measured on coronal (transaxial) MRIs at equal Z-axis intervals (156 μm) using SliceOmatic software v4.3 (TomoVision, Montreal, Canada) as previously described [49, 50]. Measurement references for FO-length and cerebral width are shown in S1 Fig. The cerebral cortex was segmented semi-automatically based on MRI contrasts and was rendered in 3D using SliceOmatic software (see Fig 1A). The 3D image was used to compute the mean cortical thickness throughout the cerebral hemisphere using Amira v5.2 (Visage Imaging, San Diego, CA, USA) as previously described [8, 49, 50].

## Gyrification index

Gyrification index (GI) values were measured using SliceOmatic software (TomoVision) as previously described [13]. Briefly, the cortical outer contour and perimeters of each sulcus were separately computed from coronal MRIs. The ratio between the outer contours and sum of sulcal perimeters was calculated for all MRIs, and the mean values were used for the global-GI. Furthermore, sulcal-GI was determined as the mean value from seven primary sulci using the same contour-to-perimeter ratio from MRIs.

## Rostrocaudal GI distribution

Rostrocaudal GI distribution was mapped using SliceOmatic software (TomoVision) as previously described [13]. Briefly, the inner and outer cortical contours were computed on coronal MRIs at equal z-axis intervals (156 μm). The GI for each MRI was estimated using outer-to-inner contour ratios. The coronal MRI at the anterior commissure was registered as "slice number 0" when compiling the rostrocaudal GI distribution.

## Immunohistochemical procedures

The right and left cerebral hemispheres were separated at the longitudinal cerebral fissure and then immersed overnight in 30% sucrose in 10 mM phosphate-buffered saline at pH 7.4. The cerebral hemispheres were embedded in optimal cutting temperature compound and serially sectioned at intervals of 100 μm in the coronal plane by a Retratome (REM-700; Yamato Koki Industrial Co., Ltd., Asaka, Japan) with a refrigeration unit (Electro Freeze MC-802A, Yamato Koki Industrial). The sections were serially collected in vials containing 4% PFA solution. Four sets of coronal sections from the cerebral hemisphere were selected by referencing cortical laminar structures in each region [51].

All immunohistochemical procedures were performed on floating sections as previously described [52], with slight modifications. To enable comparison of immunostaining results, all

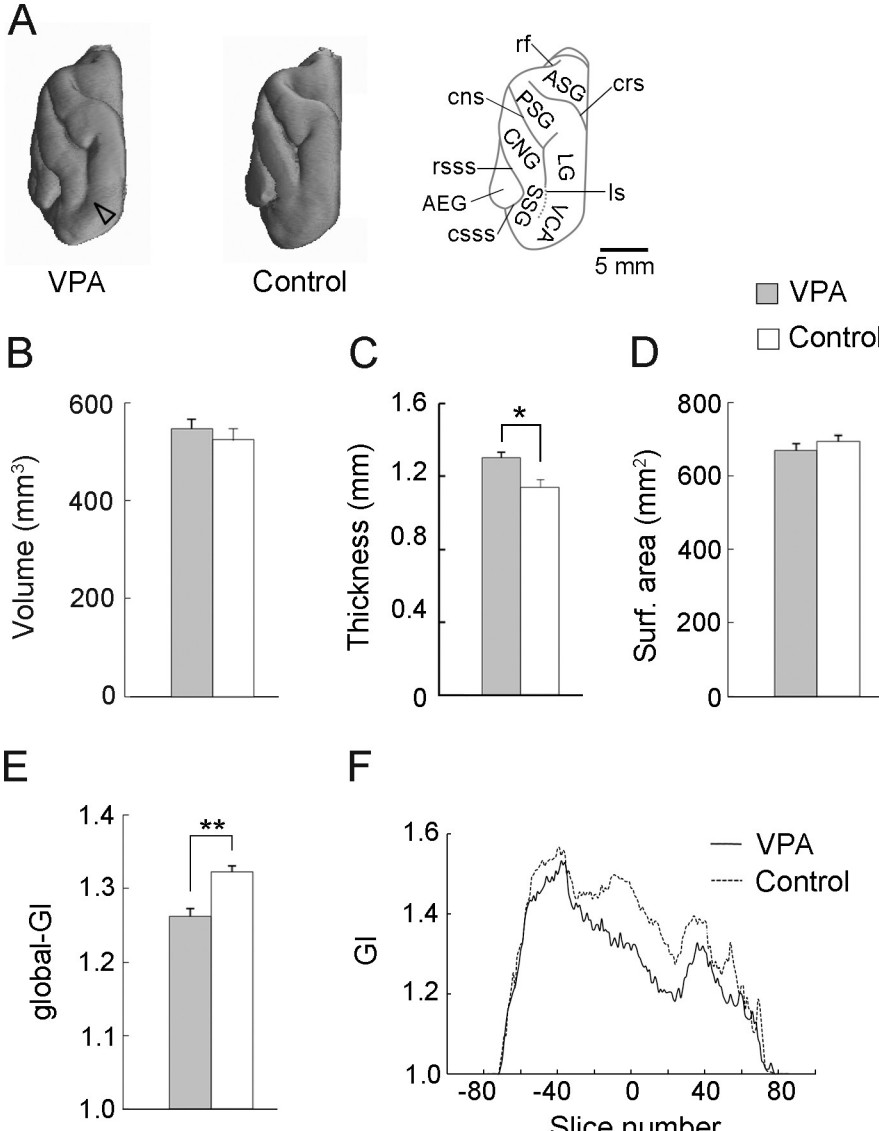

**Fig 1. Developmental VPA treatment alters some aspects of cortical development at PD 20.** (A) 3D-rendered images of the left cerebral hemisphere, dorsal view, for VPA-treated (left) and control (center) ferrets. An illustration of the dorsal surface of the left hemisphere indicating primary gyri and sulci (right). (B) Cortical volume. (C) Mean cortical thickness throughout the cerebral hemisphere. The values were computed from reconstructed 3D cortices generated from semi-automatically segmented images based on 3D MRIs. (D) Cortical surface area. (E) Global-gyrification index (global-GI). (F) Rostrocaudal distribution of the GI throughout the cerebral hemisphere. Compared to controls, lower GI values were observed in VPA-treated ferrets in the intermediate region around the anterior commissure on the rostrocaudal axis. Data are shown as mean ± SEM. Significance is indicated for Student's $t$-tests at $^{*}P < 0.01$, $^{**}P < 0.001$, number of cerebral hemispheres = 8. AEG, anterior ectosylvian gyrus; ASG, anterior sigmoid gyrus; CNG, coronal gyrus; cns, coronal sulcus; crs, cruciate sulcus; csss, caudal suprasylvian; LG, lateral gyrus; ls lateral sulcus; PSG, posterior sigmoid gyrus; rsss, rostral suprasylvian sulcus; SSG, suprasylvian gyrus; VCA, visual cortical area.

sections were processed under identical staining conditions. The same set of solutions was used for the antigen retrieval treatment, including the primary and secondary antibodies, with consistent times, temperatures, and number of PBS washings. The following primary antibodies produced highly specific immunostaining in ferret tissues in previous studies: a rat antibody to BrdU (1:1,000; ab6326, Abcam, Cambridge, UK) [52], a rabbit antibody to Olig2

(1:1,000; IBL, Gunma, Japan) [53], and mouse antibodies to GFAP (1:1,000, G 3893, Sigma-Aldrich) [54] and parvalbumin (PV; 1:500; PV235, Swant, Switzerland) [52, 54]. Anti-PV was used as a marker for bRG-derived neurons in ferret cortex on PD 20, as reported previously [52]. The sections were subsequently incubated in appropriate secondary antibodies, including Alexa 488 donkey anti-mouse IgG (1:500; A21202 Thermo Fisher Scientific, Waltham, MA, USA), Alexa 555 donkey anti-rabbit IgG (1:500; A31572, Thermo Fisher Scientific), Alexa 555 donkey anti-mouse IgG (1:500; A31570, Thermo Fisher Scientific), and Alexa 647 chicken anti-rat IgG (1:500; A-21472, Thermo Fisher Scientific). Sections were then stained using Hoechst.

## Evaluating immuno- and BrdU-labeled cell density

Immuno- and BrdU-labeled section images were acquired by serial digital sectioning (10 sections at 1 μm plane thickness) using an Axio Imager M2 ApoTome.2 microscope with a 20× objective equipped with an AxioCam MRm camera (Zeiss, Gottingen, Germany) with Zen 2.3 blue edition software (Zeiss). Immuno- and BrdU-labeled cell density was estimated by the disector method using systematic random sampling in accordance with a previously published work [52]. Counting frames with three or six square boxes (box size = 80 μm × 80 μm) were used to systematically select the region of interest (ROI) randomly superimposed on the outer stratum (OS; layers II and III) and inner stratum (IS; layers IV to VI) of gyral crowns and sulcal floors. The percentage of immuno- or BrdU-labeled cells was calculated by summing the cells counted within all ROIs from eight cerebral hemispheres of VPA-treated and control ferret pups.

## Cortical thickness in sulcal floors and gyral crowns

Thickness of the entire depth and three cortical strata, i.e., layer I, OS and IS, were measured at the primary sulci floors and the primary gyri crowns from imaged Hoechst-stained sections using ImageJ Software (NIH, Bethesda). A Mean for each sulcal or gyral region was calculated from one point on five serial images at 1-μm intervals on the z-axis. The ratio of the IS thickness to the OS thickness (OS/IS ratio) was calculated to evaluate the mechanical forces associated with cortical folding and infolding.

## Statistical analysis

Measurements from the left and right hemispheres were quantified separately. A paired sampled $t$-test demonstrated no significant differences between the hemispheres, so data from each hemisphere were considered to be independent samples. Student's $t$-tests were used to statistically evaluate the body weight, brain weight, cortical volume, FO-length of the cerebral hemisphere, mean cortical thickness, global-GI, and cortical surface areas between VPA-treated and control ferrets. Significant region-specific cerebral width differences were evaluated using a repeated measures two-way ANOVA with region and treatment as factors. Gyrus- or sulcus-specific differences in cortical thickness, OS/IS ratio, and immuno- and BrdU-labeled cell density were also evaluated using repeated measures two-way ANOVAs with gyrus/sulcus and treatment as factors. Scheffe's test was used for post-hoc testing when two-way repeated measures ANOVAs revealed significant interactions and simple main effects at $P < 0.05$.

## Results

### MRI-based morphometry

3D-rendered images of the dorsal surface of the left cerebral hemisphere from VPA-treated and control ferrets at PD 20 are shown in Fig 1A. The posterior sigmoid and central gyri were

optically expanded in VPA-treated ferrets compared to those in control ferrets. Although there was no significant difference in cerebral hemisphere FO length between VPA-treated and control ferrets (S1 Fig), repeated measures two-way ANOVA revealed a significant effect of region [$F_{(2,28)}$ = 121.926; $P < 0.001$] and a region × treatment interaction [$F_{(1,14)}$ = 3.9492; $P < 0.05$] on cerebral width and Scheffe's test revealed a significantly smaller cerebral width at the level of the posterior commissure ($P < 0.001$) (S1 Fig). Therefore, the altered cerebral hemisphere shape in VPA-treated ferrets (Fig 1A) was morphometrically characterized by narrower cerebral width at the caudal region than in control ferrets. Notably, no differences in cortical volume were observed between VPA-treated and control ferrets (Fig 1B), even though the mean cortical thickness was greater throughout the cerebral hemisphere in VPA-treated ferrets than in control ferrets ($P < 0.01$) (Fig 1C).

Cortical surface area also did not differ between VPA-treated and control ferrets (Fig 1D). However, the global-GI was significantly lower in VPA-treated ferrets (1.265 ± 0.027) compared to control ferrets (1.327 ± 0.018) ($P < 0.001$; Fig 1E). The rostrocaudal GI distribution map revealed that, compared to control ferrets, VPA-treated ferrets had lower GI values in an intermediate region around the anterior commissure on the rostrocaudal axis (Fig 1F). This area largely covers the frontal region.

Surface area and sulcal-GI were independently assessed for each primary sulcus to identify those affected by the neonatal VPA treatment. Repeated measures two-way ANOVA revealed a significant main effect of sulcus [$F_{(9,126)}$ = 581.260; $P < 0.001$] and sulcus × treatment interaction [$F_{(9,126)}$ = 15.031; $P < 0.001$] on the sulcal surface area and a significant main effect of sulcus [$F_{(9,126)}$ = 719.075; $P < 0.001$], treatment [$F_{(1,14)}$ = 19.876; $P < 0.001$], and sulcus × treatment interaction [$F_{(9,126)}$ = 24.452; $P < 0.05$] on sulcal GI. Compared to control ferrets, VPA-treated ferrets had a significantly reduced sulcal surface area in the rostral suprasylvian sulcus (rsss) ($P < 0.001$) and splenial sulcus (ss) ($P < 0.001$), but a larger sulcal area in the lateral sulcus (ls) [$P < 0.001$; Fig 2A]. Reductions in sulcal surface area were accompanied by significantly smaller sulcal-GI in the rsss ($P < 0.001$) and ss ($P < 0.001$) (Fig 2B) in VPA-treated ferrets compared to controls. Optical observations on MR images indicated that these two sulci were rostrocaudally extended through the intermediate regions of the cerebral cortex and appeared shallower in VPA-treated ferrets than in control ferrets (Fig 2C). Repeated measures two-way ANOVA revealed a significant effect of sulcus [$F_{(6,84)}$ = 121.926; $P < 0.001$], treatment [$F_{(1,14)}$ = 121.926; $P < 0.01$], and sulcus × treatment interaction [$F_{(6,84)}$ = 3.9492; $P < 0.001$] on sulcal floor cortical thickness. Notably, the thickness of rsss and ss cortical floors was a significantly greater in the VPA-treated ferrets than in control ferrets ($P < 0.001$; S2 Fig). Compared to controls, VPA-treated ferrets also showed greater sulcal-GI of the presylvian sulcus (prs) ($P < 0.001$; Fig 2B), with no alterations in the sulcal surface area (Fig 2A) or the cortical thickness of the sulcal floor (S2 Fig). Although no significant difference was detected in the ls sulcal-GI (Fig 2B), the ls was more infolded in VPA-treated ferrets than in control ferrets. Optical observation of 3D-rendered images (arrowhead in Fig 1A) and coronal MRIs (Fig 2C) indicated that increased infolding was prevalent in the caudal region. This change was reflected by the results demonstrating increased sulcal surface areas in VPA-treated ferrets (Fig 2A).

## Histocytological analysis

A recent report indicated that a majority of PV-positive neurons in the ferret cortex during PD 20 originate from bRGs [52]. PV-positive neuron density and cortical thickness of the primary sulcus floors and the primary gyrus crowns were assessed in VPA-treated and control ferrets on PD 20 (Fig 3; S3 and S4 Figs). Repeated measures two-way ANOVA revealed a

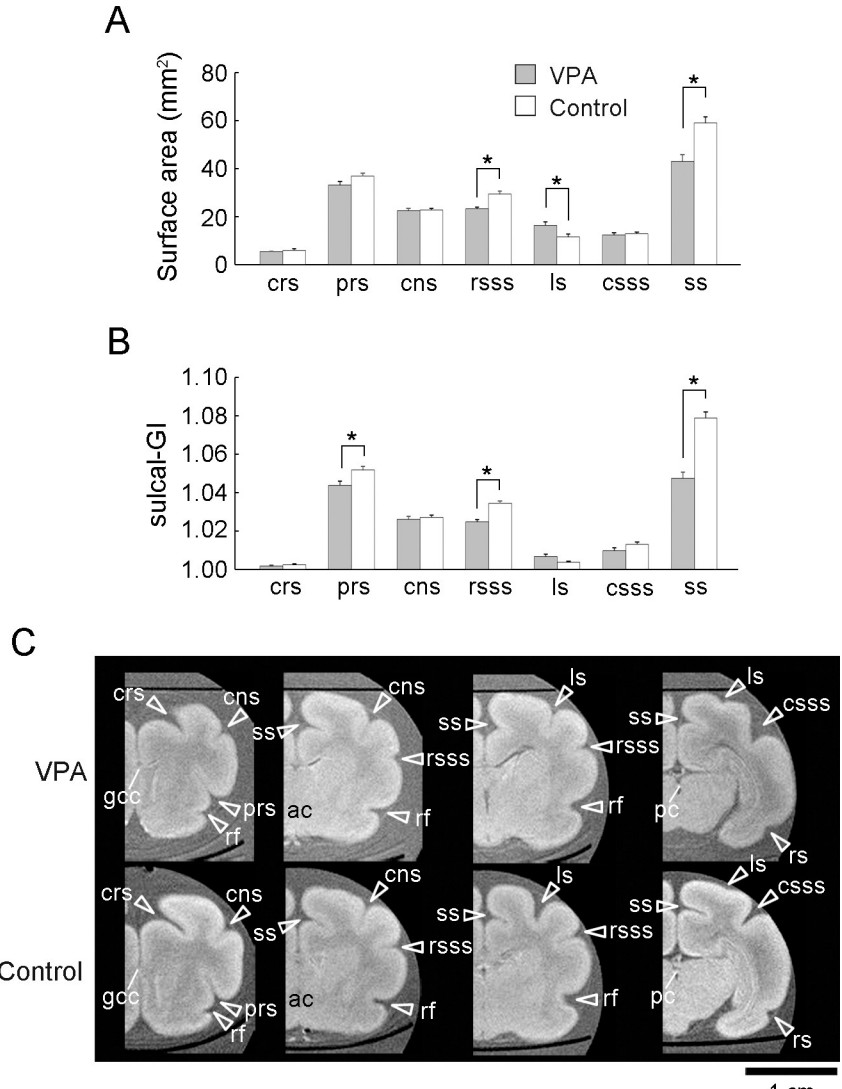

**Fig 2. Sulcal surface areas, sulcal-GI, and *ex vivo* MRIs of the cerebrum in the coronal plane from VPA-treated and control ferrets at PD 20.** (A) Bar graphs of the sulcal surface areas. (B) Sulcal gyrification index (sulcal-GI). (C) *Ex vivo* cerebrum MRIs in the coronal plane generated using RARE sequence with short TR and minimum TE settings. Coronal MRIs are shown from the left to right in the following order: the prefrontal region at the rostral end of the genu of the corpus callosum (gcc), the frontal region at the anterior commissure (ac), the parietotemporal region at the caudal end of the rhinal fissure, and the parieto-occipital region at the posterior commissure (pc). Data are shown as mean ± SEM. Significance is indicated using Scheffe's test at *$P < 0.001$; number of cerebral hemispheres = 8. cns, coronal sulcus; crs, cruciate sulcus; csss, caudal suprasylvian; ls, lateral sulcus; prs, presylvian sulcus; rsss, rostral suprasylvian sulcus; ss, splenial sulcus.

significant main effect of treatment on PV-positive neuron density in rsss [$F_{(1,14)} = 9.909$; $P < 0.01$], ls [$F_{(1,14)} = 7.417$; $P < 0.05$], and ss [$F_{(1,14)} = 12.644$; $P < 0.01$]. Scheffe's tests revealed significantly increased PV-positive neuron density in both the IS and OS of the rsss, ss, and ss floors in VPA-treated ferrets (Fig 4A). Simultaneous BrdU and VPA injections on PD 7 revealed that some PV-positive neurons were labeled through the IS and OS of the cortex on PD 20 (Fig 3). Repeated measures two-way ANOVA revealed a significant main effect of treatment on PV-positive neuron density in rsss [$F_{(1,14)} = 9.909$; $P < 0.01$], ls [$F_{(1,14)} = 7.417$; $P < 0.05$], and ss [$F_{(1,14)} = 12.644$; $P < 0.01$]. PV-positive/BrdU-labeled neuron density was

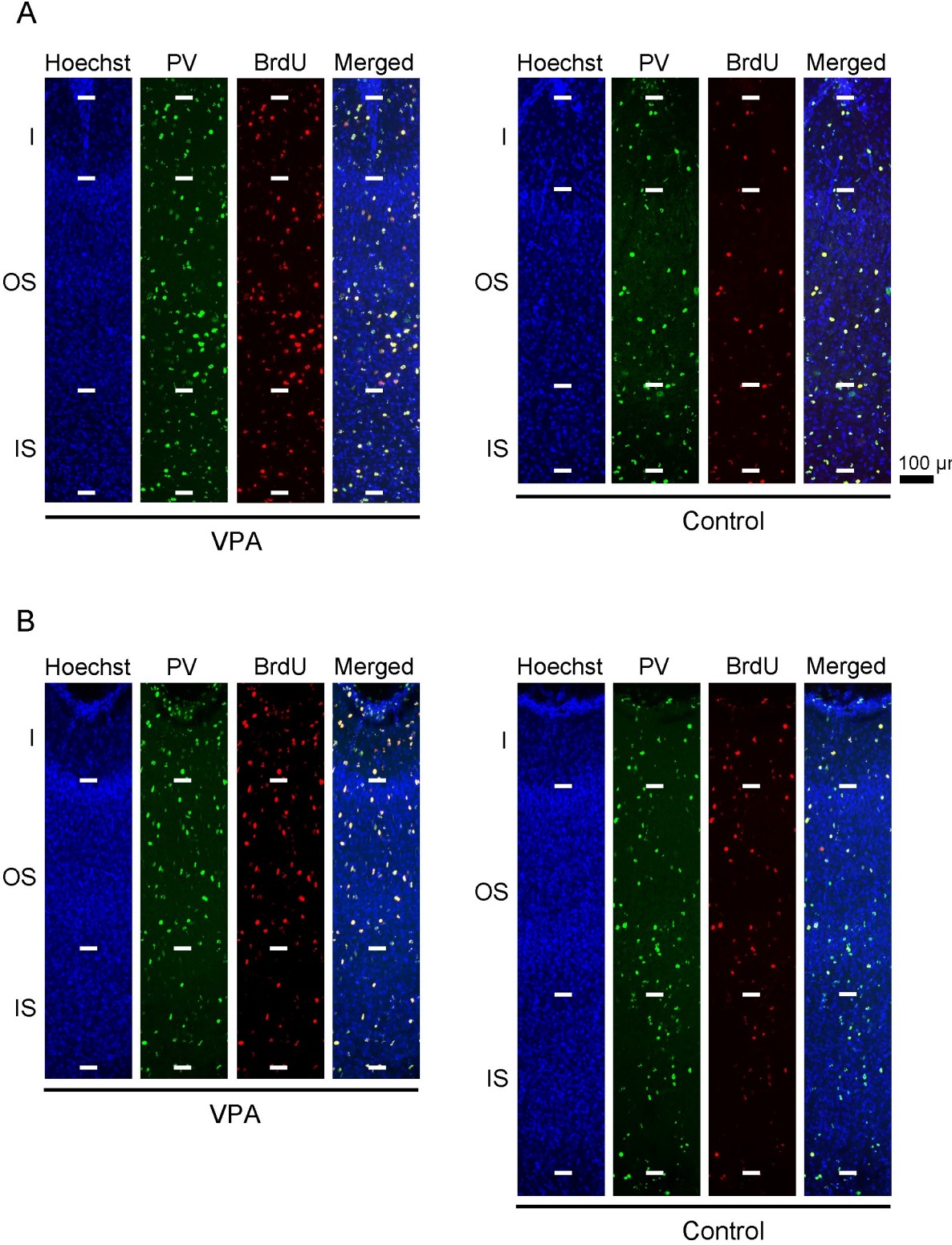

**Fig 3. Parvalbumin neuron immunofluorescence with BrdU labeling and Hoechst staining in sulcal floors of the cerebral cortex of VPA-treated and control ferrets at PD 20.** (A) Cortical depth of the rostral sylvian sulcus (rsss) floor. (B) Cortical depth of the lateral sulcus (ls) floor. The rsss and ls were selected as representative sulci with cortical floors that were thickened or thinned by neonatal VPA exposure, respectively. PV, parvalbumin.

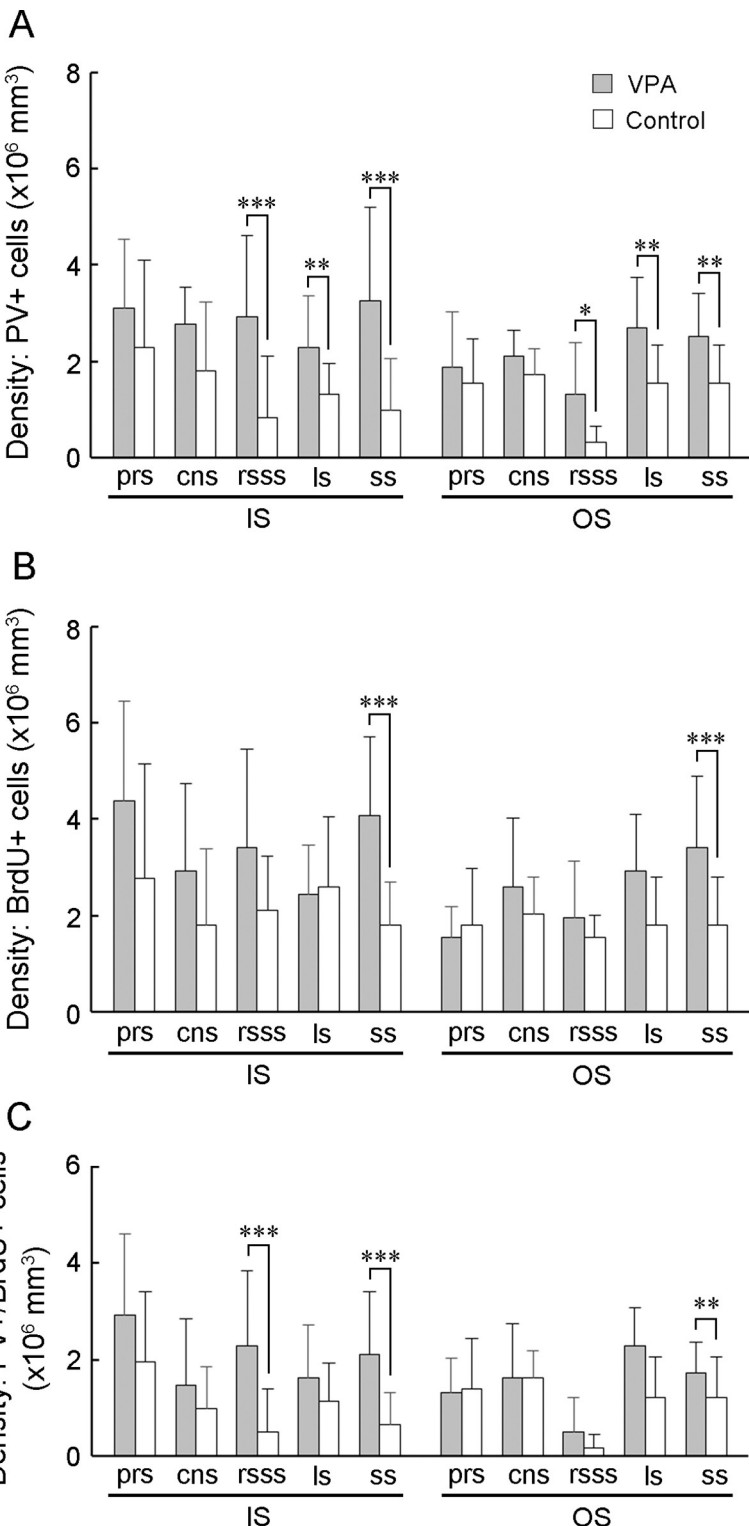

**Fig 4. PV-positive, BrdU-labeled, and PV-positive/BrdU-labeled cell density in the sulcal floors of the cerebral cortex of VPA-treated and control ferrets at PD 20.** (A) PV-positive neuron density. (B) PV-positive/BrdU-labeled cell density. (C) BrdU-labeled cell density. Data are shown as mean ± SEM. Significance is indicated using Scheffe's test at * $P < 0.05$, ** $P < 0.01$, or *** $P < 0.001$; number of cerebral hemispheres = 8. cns, coronal sulcus; IS, inner stratum; ls, lateral sulcus; rsss, rostral suprasylvian sulcus, ss, splenial sulcus; OS, outer stratum; prs, presylvian sulcus.

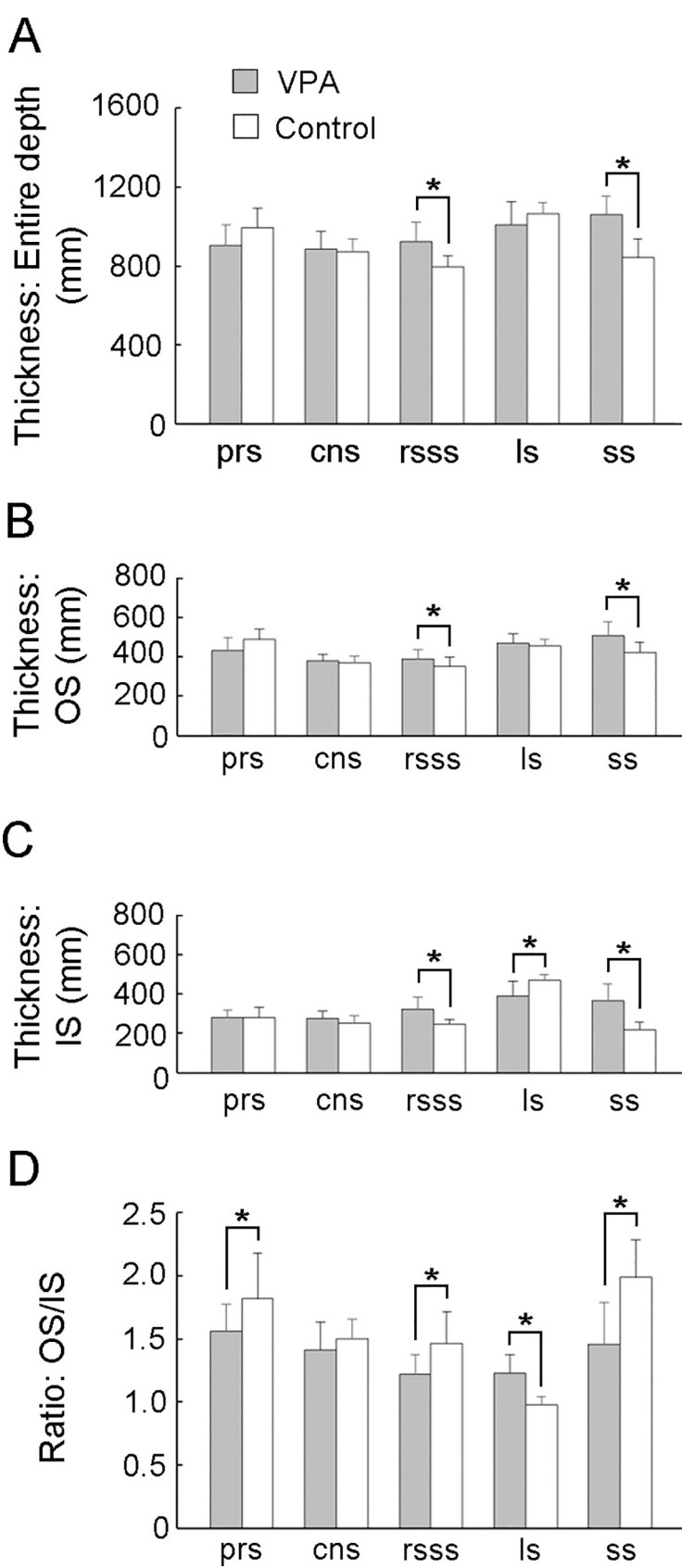

**Fig 5. Cortical layer thickness and Inner Stratum (IS) thickness to Outer Stratum (OS) thickness ratio (IS/OS) in sulcal floors of the cerebral cortex of VPA-treated and control ferrets at PD 20.** (A) Thickness of the entire cortical depth. (B) OS thickness. (C) IS thickness. (D) IS/OS ratio. Data are shown as mean ± SEM. Significance is indicated using Scheffe's test at * $P < 0.05$ or ** $P < 0.001$; number of cerebral hemispheres = 8. cns, coronal sulcus; IS, inner stratum; ls, lateral sulcus; rsss, rostral suprasylvian sulcus; ss, splenial sulcus; OS, outer stratum; prs, presylvian sulcus.

significantly greater in the IS of the rsss floors and in both strata of the ss floors in VPA-treated ferrets than in control ferrets (Fig 4C).

Next, the percentage of BrdU-labeled cells in the cortical floor of the rsss and coronal sulcus (cns) that were PV- and Olig2-positive was assessed. These two sulcal regions were selected because significantly increased PV-positive/BrdU-labeled neuron density was observed in the rsss floors, but not in the cns floors, of VPA-treated ferrets compared to controls (Fig 4B and 4C). Control ferrets showed a variable percentage of PV-positive/BrdU-labeled cells across sulcal regions (cns floors = 72.2%; rsss floors = 15.6%; S2 Table). The percentage of PV-positive/BrdU-labeled cells in the rsss was 44.4% in VPA-treated ferrets, which was significantly higher than that in control ferrets (S2 Table). While BrdU-labeling was found in some Olig2-positive cells (S5 Fig), VPA treatment did not alter the percentage of Olig2-positive/BrdU-labeled cells in these two sulci (S3 Table). In addition, Olig2-positive/BrdU-labeled cell density did not differ between VPA-treated and control ferrets in the rsss and cns floors (S4 Table). Thus, a higher proportion of SVZ progenitor differentiation into PV-positive neurons was involved in the increased density of those neurons in particular sulcal regions.

Repeated measures two-way ANOVA revealed a significant effect of treatment × sulcus interaction on rsss and ss floor cortical thickness [$F_{(4,56)} = 10.319$ $P < 0.001$] and the entire cortical depth. Scheffe's test indicated a significantly thicker cortex of the rsss and ss floors in VPA-treated ferrets than in control ferrets (Fig 5A). Repeated measures two-way ANOVA revealed a significant effect of treatment in rsss [$F_{(1,14)} = 9.574$; $P < 0.01$] and of the treatment × stratum interaction in ls [$F_{(2,28)} = 5.107$; $P < 0.05$] and ss [$F_{(2,28)} = 8.410$; $P < 0.01$] on IS and OS thickness. Scheffe's test indicated a significantly thicker IS of rsss and ss, thicker OS of ss, and thinner IS of ls in VPA-treated ferrets than in control ferrets (Fig 5B and 5C).

The OS/IS ratio was calculated to evaluate the mechanical forces associated with cortical folding and infolding. Repeated measures two-way ANOVA revealed a significant treatment × sulcus interaction in sulcal floors [$F_{(4,56)} = 8.331$; $P < 0.05$] on the OS/IS ratio. In Scheffe's test, the OS/IS ratio in the rsss, ss, and prs floors was significantly smaller in VPA-treated ferrets than in control ferrets (Fig 5D). These results indicate that the IS was relatively expanded in these three sulcal floors. Conversely, a significantly thinner IS and greater OS/IS ratio in VPA-treated ferrets than in control ferrets were observed in the ls floors (Fig 5C and 5D), indicating a relative expansion of the OS.

In gyral regions, repeated measures two-way ANOVA revealed a significant effect of treatment on PV-positive neuron density in the anterior sigmoid gyrus (ASG) [$F_{(1,14)} = 4.900$; $P < 0.05$] and the suprasylvian gyrus (SSG) [($F_{(1,14)} = 5.305$; $P < 0.05$). VPA-treated ferrets showed a significantly greater PV-positive neuron density in the IS of the ASG and OS of the SSG in Scheffe's test (Fig 6A). BrdU-labeled cell density and PV-positive/BrdU-labeled neuron density did not differ between VPA-treated and control ferrets in other examined gyral crowns (Fig 6B and 6C). In gyral crown cortical thickness, repeated measures two-way ANOVA revealed a significant effect of treatment [$F_{(1,14)} = 6.492$; $P < 0.05$] and Scheffe's test revealed a significantly greater entire cortical depth in ASG of VPA-treated ferrets than in that of control ferrets (Fig 7A). There were no differences in IS and OS thicknesses or the OS/IS ratio between VPA-treated and control ferrets in any examined gyri (Fig 7B–7D).

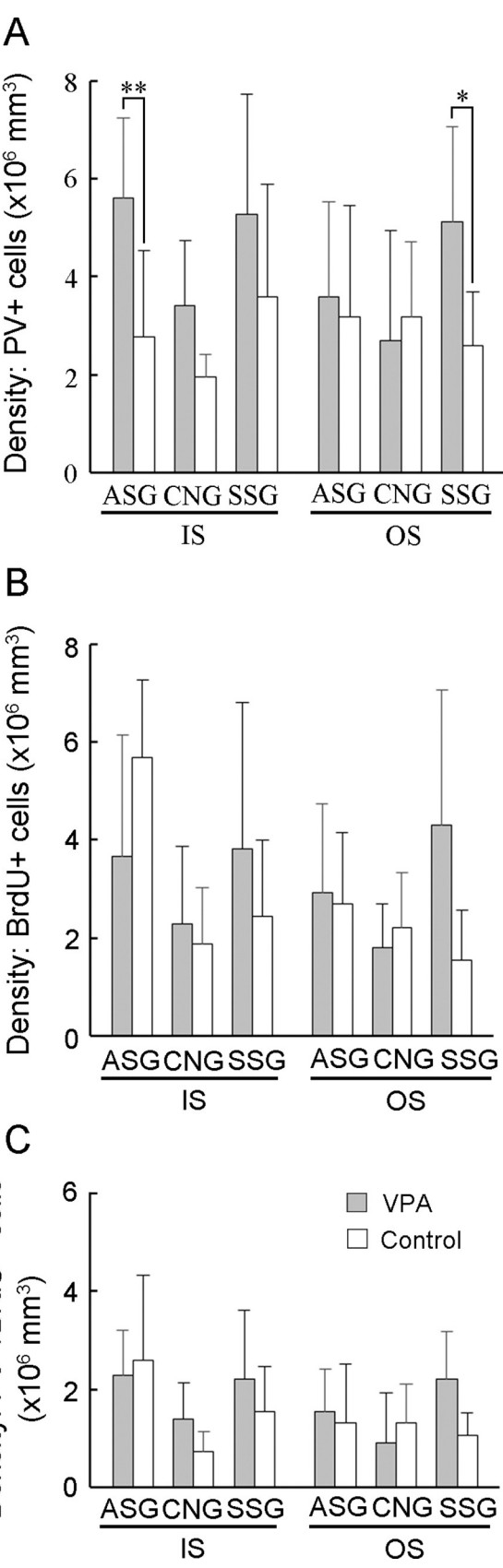

**Fig 6. PV-positive neuron density, BrdU-labeled cell density, and PV-positive/BrdU-labeled cell density in gyral crowns of the cerebral cortex of VPA-treated and control ferrets at PD 20.** (A) PV-positive neuron density. (B) PV-positive/BrdU-labeled neuron density. (C) BrdU-labeled cell density. Data are shown as mean ± SEM. Significance is indicated using Scheffe's test at * $P < 0.01$ or ** $P < 0.001$; number of cerebral hemispheres = 8. ASG, anterior sigmoid gyrus; CNG, coronal gyrus; IS, inner stratum; OS, outer stratum; SSG, suprasylvian gyrus.

## Sulcal infolding in VPA-treated ferrets

Previous studies observing the laminar structure of the cerebral cortex have revealed tangential compressions of the sulcal OS and the gyral IS, with vertical expansions of the reciprocal strata in both regions [51, 55]. The current study examined possible factors that influence the vertical expansion of each cortical stratum in sulcal floors (Fig 8A). VPA-treated ferrets showed reduced sulcal surface areas and sulcal-GI with thickened floor cortices in the rsss and ss compared to control ferrets. Notably, the OS/IS ratio was reduced in these sulcal floors, indicating vertical expansion of the IS. Compared to control ferrets, VPA-treated ferrets showed increased densities of PV-positive neurons and PV-positive/BrdU-labeled cells in the IS of rsss and ss. Therefore, increased PV-positive neuron density with IS thickening of the floor cortex in the rsss and ss may enhance vertical expansion, resulting in reduced sulcal infolding in VPA-treated ferrets compared to control ferrets. (Fig 8B).

Sulcal GI was also reduced in the prs of VPA-treated ferrets compared to that in the prs of control ferrets. A significantly decreased OS/IS ratio in the prs floor cortex was marked in VPA-treated ferrets compared to that in control ferrets. However, the cause of this relative IS expansion was not clear at the histocytological level because VPA treatment did not induce changes in PV-positive, BrdU-positive, or PV-positive/BrdU-labeled neuron densities in the prs compared to control ferrets.

In contrast to the rsss and ss, the sulcal surface area and sulcal GI were increased in the ls of VPA-treated ferrets compared to those in control ferrets. The OS/IS ratio in the ls revealed a relative OS expansion in VPA-treated ferrets. Compared to control ferrets, VPA-treated ferrets showed increased PV-positive neuron density and BrdU-labeled cell density in the OS of ls. Therefore, increased PV-positive neuron density with OS thickening of the ls floors may reduce vertical expansion, resulting in enhanced sulcal infolding of the ls in VPA-treated ferrets (Fig 8C).

Histocytometrical findings in the cns, which did not display alter sulcal infolding, are shown for reference (Fig 8A). There were no changes in the cortical thickness, OS/IS ratio, or PV-positive, BrdU-positive, and PV-positive/BrdU-labeled neuron density in the cortical strata of the cns floors.

In gyral regions, PV-positive neuron density was increased in the ASG and SSG crowns of VPA-treated ferrets compared to that in the ASG and SSG crowns of control ferrets. However, it was not clear how increased PV-positive neuron density was related to cortical folding abnormalities. VPA treatment did not alter the OS/IS ratios for the ASG and SSG crowns, even though ASG crown thickening was observed. Thus, the relationship between increased neuron densities in mechanical forces of cortical folding was ambiguous in gyral regions.

## Discussion

Developmental VPA exposure can induce ASD-like behavioral phenotypes, including reduced social interaction in mice [35, 56], rats [36–38], common marmosets [57], and ferrets [45]. VPA-induced behavioral deficits might be associated with cortical dysgenesis characterized by increasing neuron numbers and cortical thickening in humans [39], mice [41, 43], and rats [44]. Many studies have used VPA exposure during the equivalent of the second trimester as

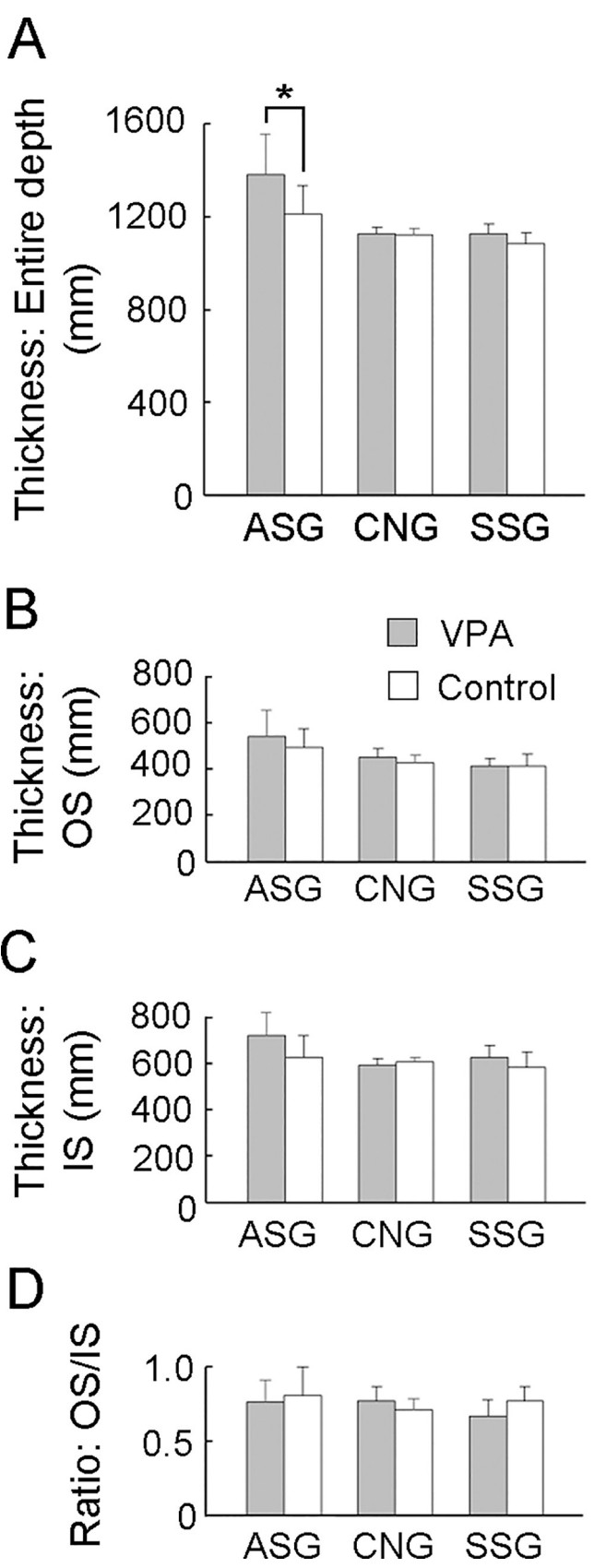

**Fig 7. Cortical layer thickness and Inner Stratum (IS) thickness to Outer Stratum (OS) thickness ratio (IS/OS) in gyral crowns of the cerebral cortex of VPA-treated and control ferrets at PD 20.** (A) Thickness of the entire cortical depth. (C) OS thickness. (D) IS thickness. (E) IS/OS ratio. Data are shown as mean ± SEM. Significance is indicated using Scheffe's test at * $P < 0.01$; number of cerebral hemispheres = 8. ASG, anterior sigmoid gyrus; CNG, coronal gyrus; IS, inner stratum; OS, outer stratum; SSG, suprasylvian gyrus.

an animal model of ASD [36, 37, 40, 44, 57]. However, VPA exposure during early postnatal life can also induce ASD -like behavioral phenotypes in mice [35, 56] and ferrets [45]. These studies indicate that VPA exposure can induce an animal model of ASD over a wide range of exposure durations, from the second trimester to infancy. The present MRI-based morphometrical study revealed that VPA exposure during the late stage of cortical neurogenesis reduced gyrification and increased cortical thickening. This is the first report to show VPA-induced altered morphology of the cerebral cortex at the macroscale using a gyrencephalic animal model.

Developmental VPA exposure is a known risk factor of ASD in humans [58]. Previous work has shown that VPA exposure during the lactation period produced an autism-like behavioral phenotype in ferret pups [45]. Therefore, we hypothesized that the current VPA exposure paradigm would produce a gyrencephalic animal model of ASD in ferrets. A number of studies have documented cerebral cortex abnormalities in humans with ASD, including region-specific cortical changes in gyrification [15–17, 19, 20, 27, 59], an increased number of neurons [28], age-dependent cortical thinning [19, 27], and regional-specific cortical thinning or thickening [17, 20, 25]. The current VPA-treated ferret model exhibited reduced gyrification with thickening of the sulcal floor cortex. Similar abnormal features in cortical morphology have been reported in a type of ASD patents [17, 20, 25].

The present study used BrdU to label newly generated cells immediately following VPA exposure and tracked the course of cortical sulcogyrogenesis in the ferret. PV immunostaining was used to track the cell origin. At PD 20, PV immunostaining is reportedly expressed in bRG-derived neurons in the ferret cortex [52]. Notably, the present study identified the primary sulci affected by neonatal VPA exposure and used histocytological methods to speculate on the cause of sulcal infolding abnormalities. Neonatal VPA exposure induced mechanical forces, which altered sulcal infolding by altering the relative expansions of the IS or OS of the sulcal floors via increased PV-positive bRG-derived neuron density. VPA is known to facilitate adult hippocampal neurogenesis [60] and embryonic stem cell differentiation into superficial cortical layer neurons in mice [61]. Conversely, *in utero* VPA exposure inhibited neurogenesis during the early stage of neocortical histogenesis in mice [41]. Thus, developmental VPA may have diverse effects that are dependent on the type of neuronal progenitor/stem cells and exposure time point. Increased density and proportion of newly generated (BrdU-labeled) PV-positive bRG-derived neurons at PD 20 in VPA-treated ferrets suggest that VPA can induce self-renewal and neurogenesis of bRG cells.

The present findings further reveal that neonatal VPA-induced cortical region-specific sulcal infolding abnormalities. These diverse VPA effects may be explained in part by the timing of VPA exposure. The duration of VPA administration (on PDs 6 and 7) corresponds to the development of early-generated primary sulci, such as the rsss and ss [12, 13]. By contrast, the ls belongs to the late-generated primary sulcus, which does not emerge and develop until after PD 10 [12, 13]. The ls was infolded in the pariet al., region, caudodorsal to the rsss. VPA may facilitate bRG-derived production of IS neurons in developing primary sulci and OS neurons prior to sulcal development. Notably, VPA exposure did not alter bRG-derived neurogenesis and sulcal infolding in the cns even though it is an early-generated primary sulci like the rsss and ss [12]. Previous studies have found that the cns infolded in the unimodal primary motor

## A

| | | | prs | cns | rsss | ls | ss |
|---|---|---|---|---|---|---|---|
| Sulcal surface area | | | ↓ | | ↓ | ⇑ | ↓ |
| Sulcal-GI | | | | | ↓ | ⇑ | ↓ |
| Thickness: Floor cortex | Entire | | | | ↑ | | ↑ |
| | OS:IS ratio | | ↓ | | ↓ | ⇑ | ↓ |
| Density | PV+ | OS | | | ⇑ | ⇑ | ⇑ |
| | | IS | | | ↑ | ↑ | ↑ |
| | BrdU+ | OS | | | | | ↑ |
| | | IS | | | | | ↑ |
| | PV+/BrdU+ | OS | | | | ⇑ | ⇑ |
| | | IS | | | | | ↑ |

→: factors for enhancing vertical expansion
⇒: factors for reducing vertical expansion

## B

Sulcal infolding ↓: rsss, ss

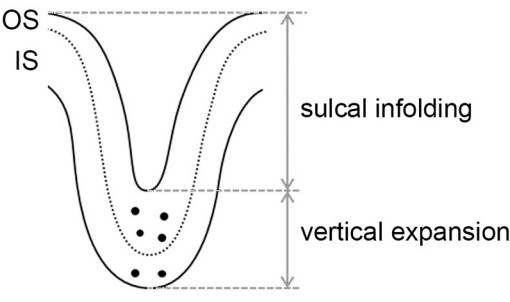

thickness: OS > IS

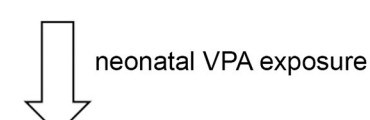
neonatal VPA exposure

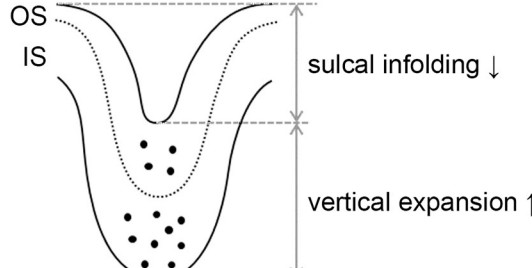

thickness: OS < IS
IS: PV-positive neuron density ↑

## C

Sulcal infolding ↑: ls

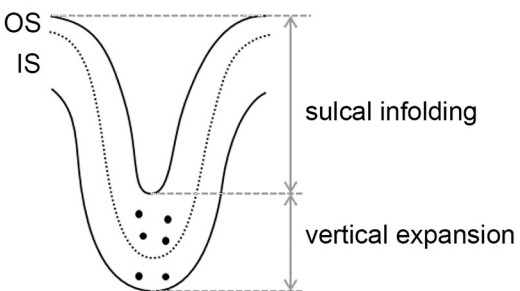

thickness: OS > IS

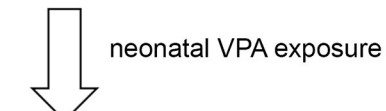
neonatal VPA exposure

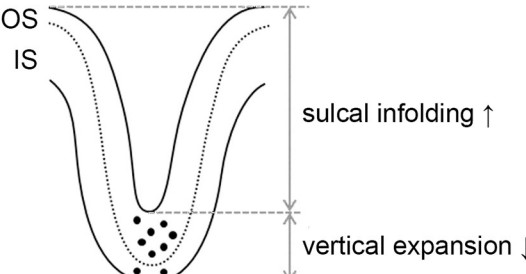

thickness: OS < IS
OS: PV-positive neuron density ↑

**Fig 8. Evidence of altered sulcal infolding in ferrets that received neonatal VPA exposure.** (A) Possible factors that influence vertical expansion of each cortical stratum in sulcal and gyral regions. (B) VPA treatment increased parvalbumin (PV)-positive neuron density and thickened the inner stratum (IS) of the rostral suprasylvian sulcus (rsss) and splenial sulcus (ss) floors. This may enhance vertical expansion, thereby reducing sulcal infolding. (C) VPA treatment increased PV-positive neuron density and thickened the outer stratum (OS) of the lateral sulcus (ls) floor. This may reduce vertical expansion, thereby enhancing sulcal infolding. cns, coronal sulcus; prs, presylvian sulcus; sulcal-GI, sulcal gyrification index.

cortex and the rsss and ss infolded in the multisensory and visual association cortices, respectively [62, 63]. Therefore, multimodal-associated cortical regions, which contain a dense number of bRG-derived neurons [52], may be vulnerable to developmental VPA exposure.

## Conclusions

The present findings revealed that VPA exposure during the late stage of neurogenesis reduced gyrification by thickening the sulcal floor cortex, particularly in the frontal and parietotemporal divisions. Histocytological findings indicated that developmental VPA-induced these abnormalities via mechanical compression and increased bRG-derived neuron density in the sulcal floor cortex. The current VPA-treated ferret model produced abnormal cortical morphology features present in a type of human patients with ASD [17, 20, 25]. The current findings can help in understanding the mechanisms of sulcogyral development and human ASD pathogenesis that is characterized by gyrification abnormalities.

## Supporting information

**S1 Fig. Fronto-Occipital [FO]-length and width of the cerebral hemisphere in VPA-treated and control ferrets at PD 20.** [A] 3D-rendered images of the left cerebral hemisphere, dorsal view, indicating measurement references for FO-length and the width of the cerebral hemisphere [VPA-treated, left; control treated, center]. Arrows indicate measurement points for the cerebral width at the genu of the corpus callosum [gcc], caudal end of the rhinal fissure [rf], and the posterior commissure [pc]. An illustration of the dorsal surface of the left hemisphere indicating the primary gyri and sulci is shown on the right. [B] FO-length of the cerebral hemisphere. [C] Width of the cerebral hemisphere. [D] Cortical volume. Data are shown as mean ± SEM. Significance is indicated using Scheffe's test at $^*$ $P < 0.001$; number of cerebral hemispheres = 8. AEG, anterior ectosylvian gyrus; ASG, anterior sigmoid gyrus; CNG, coronal gyrus; cns, coronal sulcus; crs, cruciate sulcus; csss, caudal suprasylvian; LG, lateral gyrus; ls, lateral sulcus; PSG, posterior sigmoid gyrus; rsss, rostral suprasylvian sulcus; SSG, suprasylvian sulcus; VCA, visual cortical area.
(TIF)

**S2 Fig. Cortical thickness in VPA-treated and control ferrets at PD 20.** [A] Bar graphs displaying cortical thickness of the sulcal floors. [B] Cortical thickness of the gyral crowns. Data are shown as mean ± SEM. Significance is indicated using Scheffe's test at $^*$ $P < 0.001$; number of cerebral hemispheres = 8. cns, coronal sulcus; crs, cruciate sulcus; csss, caudal suprasylvian; ls, lateral sulcus; prs, presylvian sulcus; rsss, rostral suprasylvian sulcus; ss, splenial sulcus; AEG, anterior ectosylvian gyrus; ASG, anterior sigmoid gyrus; CG, cingulate gyrus; CNG, coronal gyrus; mFC, medial frontal cortex; LG, lateral gyrus; PEG, posterior ectosylvian gyrus; PSG, posterior sigmoid gyrus; SSG, suprasylvian gyrus; VCA, visual cortical area.
(TIF)

**S3 Fig. Parvalbumin neuron immunofluorescence with BrdU labeling and Hoechst staining in sulcal floors of the cerebral cortex of VPA-treated and control ferrets at PD 20.** (A) Cortical depth of the presylvian sulcus (prs) floor. (B) Cortical depth of the coronal sulcus

(cns) floor. (C) Cortical depth of the splenial sulcus (ss) floor. PV, parvalbumin.
(TIF)

**S4 Fig. Parvalbumin neuron immunofluorescence with BrdU labeling and Hoechst staining in gyral crowns of the cerebral cortex of VPA-treated and control ferrets at PD 20.** (A) Cortical depth of the anterior sigmoid gyrus (ASG) crown. (B) Cortical depth of the coronal gyus (CNG) crown. (C) Cortical depth of the suprasylvian gyrus (SSG) crown. PV, parvalbumin.
(TIF)

**S5 Fig. Olig2 immunofluorescence with BrdU labeling and Hoechst staining in sulcal floors of the cerebral cortex of VPA-treated and control ferrets at PD 20.** (A) Cortical depth of the rostral sylvian sulcus (rsss) floor. (B) Cortical depth of the coronal sulcus (cns) floor.
(TIF)

**S1 Table. Body and brain weights of PD 20 ferrets used in the present study.**
(PDF)

**S2 Table. Percentage of PV-positive/BrdU-labeled cells in sulcal floors and gyral crowns in PD 20 ferrets.**
(PDF)

**S3 Table. Percentage of Olig2-positive/BrdU-labeled cells in the coronal and rostral suprasylvian sulci floors in PD 20 ferrets.**
(PDF)

**S4 Table. Olig2-positive cell densities with or without BrdU labeling in the coronal and rostral suprasylvian sulci floors of PD 20 ferrets.**
(PDF)

## Acknowledgments

The authors wish to thank Mr. Nobuhiro Nitta (Molecular Imaging Center, National Institute of Radiological Sciences, Chiba, Japan) for MRI measurements.

## Author Contributions

**Formal analysis:** Kazuhiko Sawada.

**Funding acquisition:** Kazuhiko Sawada, Ichio Aoki.

**Investigation:** Kazuhiko Sawada, Shiori Kamiya.

**Methodology:** Kazuhiko Sawada, Ichio Aoki.

**Supervision:** Kazuhiko Sawada.

**Validation:** Kazuhiko Sawada.

**Writing – original draft:** Kazuhiko Sawada, Ichio Aoki.

**Writing – review & editing:** Kazuhiko Sawada, Ichio Aoki.

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
