## [Decision Letter · Decision Letter 0]

24 Feb 2021

PONE-D-21-01930

Neonatal Valproic Acid Exposure Produces Altered Gyrification Related to Increased Parvalbumin-Immunopositive Neuron Density with Thickened Sulcal Floors

PLOS ONE

Dear Dr. Sawada,

Thank you for submitting your manuscript to PLOS ONE. After careful consideration, we feel that it has merit but does not fully meet PLOS ONE’s publication criteria as it currently stands. Therefore, we invite you to submit a revised version of the manuscript that addresses the points raised during the review process.

We look forward to receiving your revised manuscript.

Kind regards,

Anju Vasudevan, PhD

Academic Editor

PLOS ONE

Reviewers' comments:

**Comments to the Author**

1. Is the manuscript technically sound, and do the data support the conclusions?

Reviewer #1: Yes

Reviewer #2: Yes

Reviewer #3: Yes

2. Has the statistical analysis been performed appropriately and rigorously? 

Reviewer #1: Yes

Reviewer #2: Yes

Reviewer #3: Yes

3. Have the authors made all data underlying the findings in their manuscript fully available?

Reviewer #1: No

Reviewer #2: Yes

Reviewer #3: Yes

4. Is the manuscript presented in an intelligible fashion and written in standard English?

Reviewer #1: Yes

Reviewer #2: Yes

Reviewer #3: Yes

5. Review Comments to the Author

**Reviewer #1:** The current study showed that VPA exposure during the late stage of neurogenesis induced significantly smaller sulcal-GIs in the rostral suprasylvian sulcus and splenial sulcus but a larger lateral sulcus surface area that control ferrets. Parvalbumin-positive neuron density was significantly greater in the expanded cortical strata of sulcal floors in VPA-treated ferrets. The study is interesting, and following are my comments:

1. Although the study reported the morphological changes after VPA treatment, but the mechanisms of the phenomenon were not involved in this study.

2. According to the design of the author, BrdU was injected in P6-7, and the neurogenesis was analyzed in P20, during this period, the proliferation, differentiation and apoptosis all happened. So I don’t what was the real influence of VPA to neurogenesis, for we only saw a final results.

3. Any molecules which regulating proliferation were not explored.

4. The association between social behavior and structure of brain is also not clear.

**Reviewer #2: **The authors use ex vivo MRI-based morphometry on ferret brain. Aim was to assess whether treatment with valproic acid, a substance which is associated with autism spectrum disorder (ASD)-like behavioral phenotypes in humans that were exposed during development, would also induce cortical dysgenesis in ferrets. Abnormal gyrification is observed in humans with ASD.

There are still large gaps in knowledge on the pathophysiology of ASD. The study adds to this knowledge by demonstrating that treatment with valproic acid caused an abnormal morphological phenotype in ferrets which resembles features of aberrant morphology in ASD. The study design is straight forward, and positive aspects include the choice of an adequate model (ferrets are gyrencephalic animals), sophisticated methodology (MRI) to address the question and the ability to resist the temptation to over-interpret the data (which is commendable since it is not common these days).

There are several issues, nevertheless, that could be addressed:

1. Abstract: “however, gyrencephalic abnormalities have also not been reported”; do the authors mean: “so far, it is not known whether ferrets react to VPA treatment with gyrencephalic abnormalities”? Please clarify. Also, in the next sentence, it should presumably read “attempted to characterize”. Line 29 abstract “sulcus surface area than…”

2. Introduction, page 3, line 48: “abnormalities”, not abnormities; line 51: sulcogyrogenesis

3. Methods, line 140: What does the following sentence mean: “The following primary antibodies were reportedly produced by highly specific immunostaining in ferret tissues: …”?

4. Statistical analysis, line 176: One-way ANOVA followed by Students t-test… In the reviewer`s understanding, Students t-test is not a post hoc test which is used after ANOVA. Please clarify.

5. Results, line 250: the reference to figure 4B is not correct, should probably be 4C?

6. Results, line 264: “to reflate” means to re-start; was this what the authors meant, or should it rather read: “reflect”?

7. Generally, there are many orthographical errors throughout the manuscript.

**Reviewer #3:** This paper describes the results of potentially interesting studies on the effect of neonatal VPA exposure on gyrification abnormalities. The paper is generally well-written, the background of the study is clear, the protocols seem appropriate (however, see below), the number of animals per group is appropriate. However, the present findings need further clarification in the following points:

It would have been good to see the effects of VPA exposure on the weight of animals (and also the animal's brain), to make the study more complete.

VPA exposure during the prenatal period is more common than the neonatal period for inducing autism-like behavior. Why did you use the postnatal model?

---

## [Author Response · Author response to Decision Letter 0]

29 Mar 2021

I have addressed each of the points raised by the reviewers. Please find my point-by-point responses below. 

Reply to Comments to the Author

Point: Have the authors made all data underlying the findings in their manuscript fully available? Reviewer #1-No

Response: We added immunofluorescence images, which were not present in the original manuscript. Immunofluorescence images of the parvalbumin-positive neurons stained with BrdU and Hoechst in the floors of the presylvian sulcus (prs), coronal sulcus (cns), and splenial sulcus (ss) are shown in S3 Fig (referred in the text at line 286; marked in light green). Immunofluorescence images of the parvalbumin-positive neurons stained with BrdU and Hoechst in the crowns of the anterior sigmoid gyrus (ASG), coronal gyrus (CNG), and suprasylvian gyrus (SSG) are shown in S4 Fig (referred in the text at line 286; marked in light green). Immunofluorescence images of the Olig2-positive neurons stained with BrdU and Hoechst in the floors of the rostral sylvian sulcus (rsss) and coronal sulcus (cns) are shown in S5 Fig 5 (referred in the text at line 321; marked in light green). In addition, based on the suggestion of the reviewer, we added a table (S1 Table), which lists the body and brain weights of the VPA-treated and control ferrets (referred in the text at lines 97–99; marked in yellow).

Reply to Reviewer 1: 

#1: The current study showed that VPA exposure during the late stage of neurogenesis induced significantly smaller sulcal-GIs in the rostral suprasylvian sulcus and splenial sulcus but a larger lateral sulcus surface area that control ferrets. Parvalbumin-positive neuron density was significantly greater in the expanded cortical strata of sulcal floors in VPA-treated ferrets. The study is interesting, and following are my comments:

Points: 1) Although the study reported the morphological changes after VPA treatment, but the mechanisms of the phenomenon were not involved in this study.

2) According to the design of the author, BrdU was injected in P6-7, and the neurogenesis was analyzed in P20, during this period, the proliferation, differentiation and apoptosis all happened. So I don’t what was the real influence of VPA to neurogenesis, for we only saw a final result.

3) Any molecules which regulating proliferation were not explored.

4) The association between social behavior and structure of brain is also not clear.

Response: As the reviewer suggested, it is important to clarify the detailed mechanisms of neonatal VPA-induced morphological changes in the cerebral cortex. Specifically, how does VPA administration change SVZ progenitors/stem cells proliferation, differentiation, and apoptosis across the experimental period (PD 7 to PD 20); also, which molecules are involved in the differentiation and apoptosis of neuronal progenitors/stem cells. The same can be said for elucidating the relationship between neonatal VPA-induced abnormities in cortical morphology and social behaviors. However, the main focus of the current study is to morphometrically characterize neonatal VPA exposure-induced sulcogyrognesis of the ferret cerebral cortex on a macroscale or histological level. Our findings are the first to show developmental VPA exposure-induced morphological alterations in the cerebral cortex, which are reminiscent of gyrification abnormalities in a type of human ASD patients, using a gyrencephalic animal model. However, no direct findings have been obtained to elucidate the detailed molecular mechanisms and/or the involvement of behavioral deficits in developmental VPA-induced gyrification abnormalities. Thus, we believe that the research topics suggested by the reviewer should be considered in future studies.

Reply to Reviewer 2: 

#2: The authors use ex vivo MRI-based morphometry on ferret brain. Aim was to assess whether treatment with valproic acid, a substance which is associated with autism spectrum disorder (ASD)-like behavioral phenotypes in humans that were exposed during development, would also induce cortical dysgenesis in ferrets. Abnormal gyrification is observed in humans with ASD.

There are still large gaps in knowledge on the pathophysiology of ASD. The study adds to this knowledge by demonstrating that treatment with valproic acid caused an abnormal morphological phenotype in ferrets which resembles features of aberrant morphology in ASD. The study design is straight forward, and positive aspects include the choice of an adequate model (ferrets are gyrencephalic animals), sophisticated methodology (MRI) to address the question and the ability to resist the temptation to over-interpret the data (which is commendable since it is not common these days).

There are several issues, nevertheless, that could be addressed:

Point 1: Abstract: “however, gyrencephalic abnormalities have also not been reported”; do the authors mean: “so far, it is not known whether ferrets react to VPA treatment with gyrencephalic abnormalities”? Please clarify. Also, in the next sentence, it should presumably read “attempted to characterize”. Line 29 abstract “sulcus surface area than…”

Response: To address the points raised by reviewer 2 for the Abstract, we made the following revisions:

1) “however, gyrencephalic abnormalities have also not been reported” � “so far, it is not known whether ferrets react to developmental VPA exposure with gyrencephalic abnormalities” (lines 29–30; marked in yellow)

2) “attempted to characterize” � “characterized” (line 31; marked in light green)

3) “sulcus surface area that…” “sulcus surface area than…” (line 39; marked in yellow)

Point 2: Introduction, page 3, line 48: “abnormalities”, not abnormities; line 51: sulcogyrogenesis.

Response: The points raised by reviewer 2 regarding the Introduction are revised as follows:

1) abnormities � abnormalities (line 58; marked in yellow)

2) sucogyrogenesis � sulcogyrogenesis (line 60; marked in yellow)

Point 3: Methods, line 140: What does the following sentence mean: “The following primary antibodies were reportedly produced by highly specific immunostaining in ferret tissues: …”?

Response: The sentence is revised as follows: “The following primary antibodies produced highly specific immunostaining in ferret tissues in previous studies:” (lines 172–173; marked in yellow)

Point 4: Statistical analysis, line 176: One-way ANOVA followed by Students t-test… In the reviewer`s understanding, Students t-test is not a post hoc test which is used after ANOVA. Please clarify.

Response: Following the reviewer’s comments, “One-way ANOVAs followed by” is removed from this sentence (lines 210–213; marked in yellow).

Point 5: Results, line 250: the reference to figure 4B is not correct, should probably be 4C?

Response: We have corrected accordingly (line 297; marked in yellow).

Point 6: Results, line 264: “to reflate” means to re-start; was this what the authors meant, or should it rather read: “reflect”?

Response: Corrected: “reflated” � “revealed” (line 327; marked in yellow)

Point 7: Generally, there are many orthographical errors throughout the manuscript.

Response: I apologize for the number of orthographical errors in the manuscript. The revised manuscript has been re-checked carefully by a professional scientific editor.

Reply to Reviewer 3: 

#3: This paper describes the results of potentially interesting studies on the effect of neonatal VPA exposure on gyrification abnormalities. The paper is generally well-written, the background of the study is clear, the protocols seem appropriate (however, see below), the number of animals per group is appropriate. However, the present findings need further clarification in the following points:

Point 1: It would have been good to see the effects of VPA exposure on the weight of animals (and also the animal's brain), to make the study more complete.

Response: The body and brain weights of the VPA-treated and control ferrets are now shown in S1 Table. Also, these are mentioned in the Material and Methods section (lines 97–99; marked in yellow).

Point 2: VPA exposure during the prenatal period is more common than the neonatal period for inducing autism-like behavior. Why did you use the postnatal model?

Response: The current study focused primarily on the effect of VPA on sulcogyrogenesis of the ferret cerebral cortex, rather than inducing the ASD model. In ferrets, the early stage of sulcogyrogenesis in the cortex occurs from postnatal days 4 to 10 (Sawada and Watanabe, Congenit Anom, 2012; Sawada and Aoki, Neuroscience, 2017), which corresponds to the late stage of corticoneurogenesis (Reillo I and Borrell, Cereb Cortex, 2012). Thus, we administered VPA during PDs 6 to 7, because this is when a significant effect on the sulcogyrogenesis of the ferret cortex would be expected.

---

## [Decision Letter · Decision Letter 1]

5 Apr 2021

Neonatal Valproic Acid Exposure Produces Altered Gyrification Related to Increased Parvalbumin-Immunopositive Neuron Density with Thickened Sulcal Floors

PONE-D-21-01930R1

Dear Dr. Sawada,

We’re pleased to inform you that your manuscript has been judged scientifically suitable for publication and will be formally accepted for publication once it meets all outstanding technical requirements.

Kind regards,

Anju Vasudevan, Ph.D

Academic Editor

PLOS ONE

Reviewers' comments:

**Comments to the Author**

Reviewer #1: All comments have been addressed

Reviewer #2: All comments have been addressed

---

## [Editor Report · Acceptance letter]

12 Apr 2021

PONE-D-21-01930R1 

Neonatal Valproic Acid Exposure Produces Altered Gyrification Related to Increased Parvalbumin-Immunopositive Neuron Density with Thickened Sulcal Floors 

Dear Dr. Sawada:

I'm pleased to inform you that your manuscript has been deemed suitable for publication in PLOS ONE. Congratulations! Your manuscript is now with our production department. 

Kind regards, 

on behalf of

Dr. Anju Vasudevan 

Academic Editor

PLOS ONE